

# Knowledge and beliefs regarding cervical cancer screening and HPV vaccination among urban and rural women in León, Nicaragua

Hannah D. Rees[1], Alexandra R. Lombardo[1], Caroline G. Tangoren[1], Sara J. Meyers[1], Vishnu R. Muppala[2] and Linda M. Niccolai[3]

[1] Yale University, New Haven, CT, United States of America
[2] Department of Health Policy and Management, Yale School of Public Health, New Haven, CT, United States of America
[3] Department of Epidemiology of Microbial Disease, Yale School of Public Health, New Haven, CT, United States of America

Corresponding author
Hannah D. Rees,
hannah.rees@yale.edu

## ABSTRACT

**Background**. In Nicaragua, cervical cancer is the leading cause of cancer-related death for women ages 15–44, yet access to the HPV vaccine is limited to those with financial resources to pay for it. Cervical cytology is provided free of charge in public clinics; however, only 10% of women receive Pap smears at the nationally recommended frequency. Previous studies have not investigated how beliefs regarding cervical cancer screening may differ for urban and rural populations in Nicaragua. Furthermore, no investigation has assessed Nicaraguan women's beliefs about a potential HPV immunization campaign. Given beliefs' influence on health behavior, we investigated the structural, sociocultural, and knowledge-based factors influencing women's beliefs regarding cervical cancer screening among urban and rural women in León, Nicaragua, and assessed acceptance of a potential HPV immunization program.

**Methods**. Our sequential explanatory mixed-methods study consisted of two phases: (1) a close-ended questionnaire, followed by (2) a qualitative, in-depth interview. Our quantitative sample contained 117 urban and 112 rural participants aged 18–49. We assessed beliefs regarding cervical cancer screening using a 22-item scale, with higher scores indicating screening-promoting beliefs in simple linear and multiple linear regressions. Twenty qualitative interviews, exploring the sociocultural dimensions of knowledge and attitudes indicated by our quantitative findings, were conducted with a sample of 13 urban and 7 rural women aged 19–46.

**Results**. The multiple linear regression indicates that greater knowledge of Pap smears, HPV, and cervical cancer is significantly associated with screening-promoting beliefs after adjusting for other relevant factors. There was no significant difference in screening knowledge and beliefs for urban and rural women. Four recurrent themes representing determinants of knowledge, beliefs, and attitudes regarding cervical cancer screening arose from interviews and built on quantitative findings: (1) women's embarrassment due to the intimate nature of the Pap smear and male gender of exam provider discourages screening; (2) women believe Pap smears and cervical cancer are associated with sexual promiscuity, and this association stigmatizes women with the disease; (3) knowledge of cervical cancer prevention is limited to those who regularly attend health centers; and (4) women find screening inconvenient, believing understaffed

clinics increase patient wait time, limit time patients spend with clinicians, and delay Pap results. A fifth theme indicates (5) participants' acceptance of a potential HPV immunization program.

**Discussion**. Future interventions should focus on increasing access to information about cervical cancer prevention for women who do not regularly attend health centers. Furthermore, our results suggest that if funding were allocated to make the HPV vaccine accessible in Nicaragua, it would be well received.

## INTRODUCTION

Cervical cancer is the second most common cancer among women in developing countries, with approximately 445,000 new cases reported in less-developed regions in 2012 (*World Health Organization (WHO), 2016*). Over 80% of the burden of cervical cancer in the Americas is concentrated in Latin America and the Caribbean, with mortality rates due to cervical cancer three times higher than in the United States and Canada (*Capote Negrin, 2015*). In the Americas alone, cervical cancer takes the lives of 35,700 women each year, and is projected to increase to 51,500 annually by 2030 due to an increase in life expectancy and population growth (*Pan American Health Organization, 2014*). The vast majority of cases of cervical cancer are caused by persistent infection with specific strains of human papilloma virus (HPV) (*Wigle, Coast & Watson-Jones, 2013*); vaccination against HPV-16 and -18 can prevent nearly 70% of cases of cervical cancer (*Wigle, Coast & Watson-Jones, 2013*). In addition, screening through the use of cervical cytology is an extremely effective method to identify pre-cancerous lesions and prevent the development of cancer (*Pan American Health Organization, 2010*).

In Nicaragua, cervical cancer is the leading cause of cancer-related death for women ages 15–44, with an age-standardized mortality rate nearly double the global average (*Bruni et al., 2017*). Nicaragua has the highest incidence rate of cervical cancer in Central America and the highest mortality rate in all of Latin America, with 31 cases and 14 deaths per 100,000 women, respectively (*Bruni et al., 2017*; *Pulitzer Center, Public Radio International (PRI), 2008*). More than 900 new cases of cervical cancer are diagnosed per year in Nicaragua (*Bruni et al., 2017*). Despite the immense burden of cervical cancer in Nicaragua, access to the HPV vaccine is limited to those who have the financial resources to pay for the vaccine (*Bruni et al., 2017*). Thus, screening with cervical cytology remains the primary means of prevention (*Bruni et al., 2017*).

In Nicaragua, cervical cytology is recommended every three years following three consecutive, normal, annual Pap tests for women aged 25 to 64, though women outside of this age range are eligible to receive the exam (*Pan American Health Organization, 2010*). Cervical cytology is provided free of charge in the public sector in Nicaragua; however, only 10% of women receive Pap smears at the frequency recommended by national guidelines (*Bruni et al., 2017*; *Pan American Health Organization, 2010*). Several

studies have examined the factors that contribute to low rates of cervical cancer screening in Nicaragua and other Latin America countries. An investigation conducted in 2002 in the department of Rivas, Nicaragua, found that inadequate cervical cancer screening among women was correlated with low educational level, exclusive use of public health services, and limited knowledge of prevention and symptoms of cervical cancer (*Claeys et al., 2002*). Additionally, women with a lack of previous medical problems, who faced economic barriers, were less likely to seek out screening (*Claeys et al., 2002*). A review of five qualitative studies (*Agurto et al., 2004*) conducted in Venezuela, Ecuador, Mexico, El Salvador, and Peru indicated that barriers to cervical cancer screening included lack of access to quality health services, lack of privacy and comfort during screening, and poor service delivery. The review noted that rural women often had longer travel time to reach health care facilities and receive Pap results, and possessed a greater fear of the social acceptability of receiving a Pap smear when compared to urban women; however, no study has addressed potential differences among urban and rural women in Nicaragua (*Agurto et al., 2004*). Furthermore, past studies have addressed the sociocultural barriers to the rollout of the HPV vaccine in select low and middle-income countries (*Wigle, Coast & Watson-Jones, 2013*), though none has assessed women's beliefs about a potential HPV immunization campaign in Nicaragua—data that could be integral to the creation of a national vaccination program.

In addition to identifying structural factors limiting cervical cancer screening, the review highlighted beliefs, such as a general fear of cancer, anxiety while waiting to receive screening results, and stigma surrounding Pap smears, that contributed to low rates of preventive screening (*Agurto et al., 2004*). This finding aligns with the Health Belief Model (HBM), one of the most widely applied frameworks for health behavior (*Jones et al., 2016*). This model indicates that perceived susceptibility and severity of disease, and perceived benefits and barriers to health-promoting action, determine health behavior (*Jones et al., 2016*). Given the influence of beliefs on cervical cancer screening, past studies have drawn upon the HBM to assess beliefs as a predictor of cervical cancer screening behavior (*Austin et al., 2002*; *Burak & Meyer, 1997*; *Johnson et al., 2008*).

Accordingly, we sought to examine the structural, sociocultural, and knowledge-based factors that may influence women's beliefs regarding screening for cervical cancer in León, Nicaragua. We sought to compare these results between urban and rural women in the region, and hypothesized that urban women would hold stronger screening-promoting beliefs compared to rural women. In addition, we aimed to assess the views of both urban and rural women on the potential introduction of the HPV vaccine in the region. Our study employed a mixed methods strategy to expound upon our quantitative findings with in-depth interviews, allowing us to assess the determinants of cervical cancer screening and the underlying sociocultural factors that shape perceptions of the disease. Findings from this study can be used to guide future cervical cancer prevention efforts tailored to the needs and perspectives of urban and rural women in Nicaragua.

## MATERIALS & METHODS

### Study design and sampling strategy

The study design was sequential explanatory mixed-methods conducted over the course of eight weeks from June to August of 2016, and consisted of two phases: (1) a close-ended questionnaire administered in person by researchers for the first six weeks, followed by (2) a qualitative in-depth, face-to-face interview for the final two weeks. The quantitative component of the study was conducted first to assess potential gaps in knowledge and general beliefs regarding HPV, cervical cancer, and screening practices. It also examined acceptance of a potential HPV immunization campaign. The preliminary quantitative results influenced the creation of the open-ended discussion guide, which sought to elucidate sociocultural dimensions of knowledge and attitudes about HPV, cervical cancer, screening practices, and acceptance of HPV immunization that might further explain our quantitative findings.

Participants were recruited from three urban health centers and three rural health posts in León, the second largest city in Nicaragua (*Instituto Nacional de Información de Desarrollo (INIDE), 2012*), for both phases of the study. The health centers granted us permission to speak with women regarding participation in the study while they waited to be seen by a health care provider. We approached all adult women to participate in the study. Women eligible to participate in the quantitative component of the study were Spanish-speaking, aged 18–49, and had no history of cervical cancer or a hysterectomy. The same eligibility criteria were applied for participants in the qualitative component of the study; however, women with a history of cervical cancer or a hysterectomy were eligible to participate. As the questionnaire measured general knowledge of HPV and cervical cancer, women with a history of the disease could have potentially biased the results with knowledge gained from their specific experiences. One woman with a history of cervical cancer provided valuable insight, and was included in the qualitative sample. Women who had completed the quantitative survey were excluded from the qualitative component of the study.

### Quantitative data collection and measurements

The survey (Appendix A) was developed through the integration of validated questions from both the Cervical-Cancer-Knowledge-Prevention-64 (CCKP-64) (*Katarzyna et al., 2014*) and the Carolina HPV Immunization Measurement and Evaluation Project (CHIME) questionnaires (*McRee et al., in press*). Nicaraguan professors of medicine from National Autonomous University of Nicaragua (UNAN), León, aided in the development of the questionnaire to ensure its cultural competency and validity. The final version of the questionnaire consisted of five sections. The first section was comprised of general demographic questions. Section two covered knowledge and beliefs regarding Pap smears, as well as past experiences with the exam. The third and fourth sections assessed knowledge and beliefs regarding HPV and cervical cancer, respectively. Relevant questions were selected from CCKP-64 (*Katarzyna et al., 2014*) to assess knowledge of HPV and cervical cancer in these sections. The fifth section addressed acceptance of a potential HPV immunization campaign, utilizing questions from the CHIME questionnaire (*McRee et al., in press*). The questionnaire was piloted with 50 women at the three urban clinics.
Modifications were made to the instrument to increase the clarity and specificity of selected questions. All research procedures were approved by the Institutional Review Board at the Yale School of Medicine (HSC #1603017360) and UNAN, León.

The questionnaire was administered over a period of six weeks in health center waiting areas during their opening hours. All questionnaires were anonymous and administered by a member of the research team, who read all instrument questions and response choices to each participant after achieving informed, verbal consent. Although the sample was a convenience sample, participants were diverse in socioeconomic status, age, and urban/rural location of residence. We attempted to survey every eligible participant at the clinics each day, however we approximate that only 80% of those asked to participate completed the survey. One urban health center was larger than the others, but response rates were similar across health centers. After eliminating incomplete surveys from our sample, the final yield rate was 57% (Fig. 1). Incomplete surveys arose from participants leaving the health center before completing the questionnaire.

### Outcome variable

A belief score was created as a summation of 22 survey questions that assessed participants' beliefs regarding physical and emotional discomfort associated with Pap smears, perceived safety of the exam, the amount of time it takes to receive the exam and results, the relationship between Pap smears and sexual promiscuity, frequency of screening, comfort receiving a Pap from a male clinician, and perceived likelihood of being affected by cervical cancer in the future (Appendix B). Questions that involved four-level Likert responses (*somewhat agree, strongly agree, somewhat disagree, strongly disagree*) were dichotomized into two responses (i.e., *Pap tests are safe: 1 = agree, 0 = disagree*). Negative statements were reverse coded to ensure only screening-promoting beliefs increased one's belief score (i.e., *Pap tests are painful: 0 = agree, 1 = disagree*). The variable index was created as a simple summation of quantified answers to belief questions (range, 0–22), so that a higher belief score corresponds with screening-promoting beliefs.

### Explanatory variables

Knowledge was assessed through the creation of a knowledge index, a summation of nine questions (range, 0–9) that surveyed knowledge of Pap smears, HPV, and cervical cancer (Appendix B). A higher score corresponds to greater knowledge of these topics. Questions that involved a four-level Likert scale (*somewhat agree, strongly agree, somewhat disagree, strongly disagree*) were dichotomized into two responses (i.e., *You do not know where to receive a Pap smear: 0 = agree, 1 = disagree*). Other questions were dichotomous in nature, and coded accordingly (i.e., *Can HPV cause cervical cancer? 1 = yes, 0 = no*). The survey item that tested knowledge of the requirements of a Pap smear (*What are the requirements to have a Pap smear?*) was coded such that each correct response was awarded 0.25 points (*not menstruating, not using a vaginal douche before exam, not applying vaginal cream, medication, or suppository before exam, not having intercourse three days before exam*), while the incorrect response (*there are no requirements*) was coded to be 0.

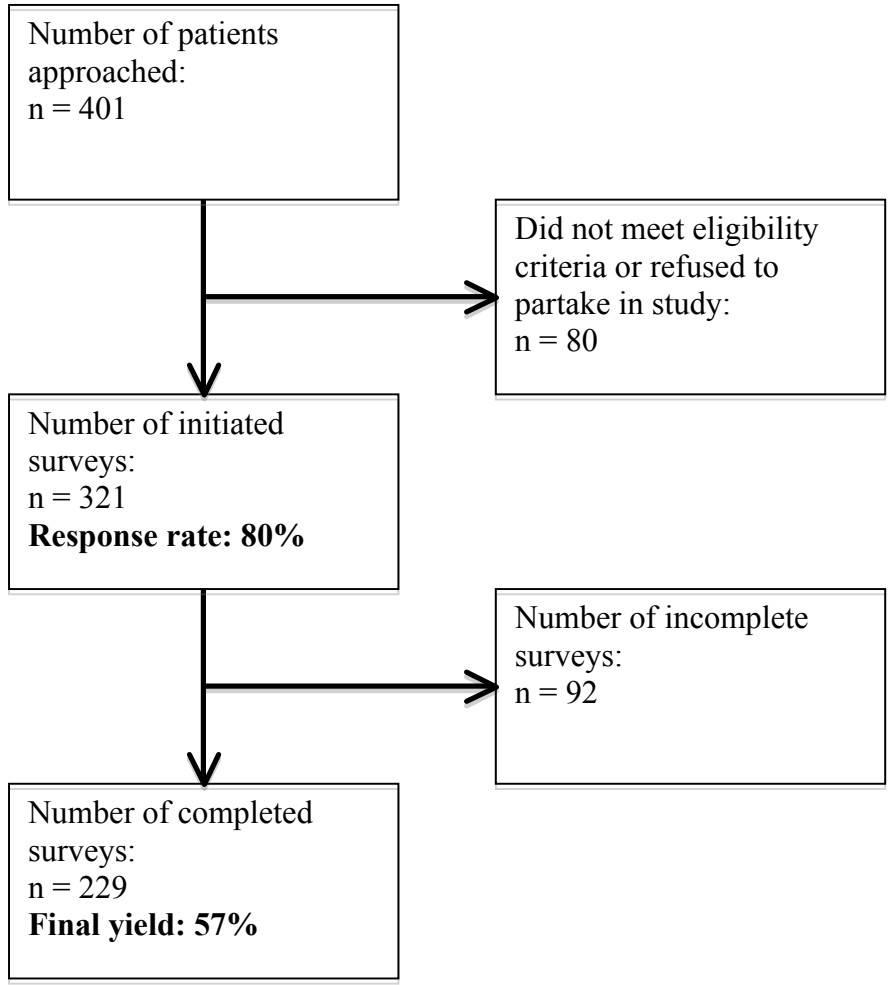

**Figure 1  Response rate and final yield of the quantitative sample.**

### Covariates

Covariates included region of residence, education, income, age, and direct contact with an individual with cervical cancer. Region of residence was a binary variable (*rural = 1, urban = 0*). Education was represented by five dummy variables, with postgraduate education being the reference level (*none or preschool, some or complete primary school, some or complete secondary school, some or complete university, postgraduate*). Income was indicated by four dummy variables, with living comfortably being the reference level (*poor, just getting by, living comfortably, rich*). Age was a continuous variable. Direct contact with an individual with cervical cancer was a binary variable. If the participant had known an individual that had been diagnosed with cervical cancer, it was coded as 1. If the participant did not know someone with cervical cancer, or reported not knowing if she knew someone, it was coded as 0.

## Quantitative data analysis

We used standard frequency analysis to describe the characteristics of the sample respondents and responses to knowledge and attitude items. To determine the correlates of beliefs, we conducted analyses using simple linear regressions and a multiple linear regression. Variables of interest from the survey (region of residence, personal contact with cervical cancer, age, knowledge of HPV and cervical cancer, education level, and income) were selected based upon theoretical knowledge of determinants of beliefs about cervical cancer screening. A correlation table was used to ensure that no two variables used in the regression had a correlation value greater than 0.4. All quantitative data were analyzed using STATA, version 14.1.

## Qualitative data collection and discussion guide

Before beginning Phase Two of the study, our advisors at UNAN, León reviewed the interview guide to ensure its clarity and cultural competency. The final discussion guide (Appendix C) contained questions addressing each participant's overall experience with health care, exposure to sexual health education, past experiences with Pap smears, personal knowledge and beliefs regarding HPV and cervical cancer, and perceived community views of the diseases. The interviews also included questions evaluating the experiences of women with HPV and cervical cancer, and how the participant makes decisions regarding her sexual health. Finally, the guide included questions to assess views on the HPV vaccine. The discussion guide was piloted with five women at urban and rural clinics to improve interview technique and identify the most effective prompts and probes.

The semi-structured interviews were completed over two weeks at each of the three urban health centers and three rural health posts during their opening hours. Interviews were conducted with women waiting to be seen by health care providers, out of earshot of other patients. The researcher provided each participant with a written and verbal description of the details of the study, and verbal consent was received from each participant. Interviews were anonymous, audio-recorded, and only accessible to the researchers. We continued to interview women until the point of theoretical saturation, i.e., when successive interviews produced no additional concepts (*Bradley, Curry & Devers, 2007*), which occurred after 21 interviews.

## Qualitative data analysis

The audio recordings were transcribed verbatim into Spanish, and were analyzed concurrently with data collection. The data were translated after analysis was completed. A deductive approach (*Bradley, Curry & Devers, 2007*) was used in the analysis of the transcripts. The interviews were first read for comprehension and a preliminary framework of codes was then applied to the data. Research team members coded three interviews independently using the preliminary code structure. Modifications to the code were made after the discrepancies in coding were resolved by negotiated consensus. This process was repeated three more times before the code structure was finalized. Each interview was coded by two researchers using Dedoose qualitative software to organize the data,

**Table 1 Quantitative sample characteristics (n = 229).**

| Demographic characteristic: | Total (n = 229) n (%) | Urban (n = 117) n (%) | Rural (n = 112) n (%) |
|---|---|---|---|
| **Mean age** | 28.1 | 28.6 | 27.6 |
| **Education** | | | |
| No formal education | 13 (5.68) | 2 (1.71) | 11 (9.82) |
| Primary school[*] | 54 (23.58) | 16 (13.68) | 38 (33.93) |
| Secondary school[*] | 105 (45.85) | 56 (47.86) | 49 (43.75) |
| University[*] | 53 (23.14) | 40 (34.19) | 13 (11.61) |
| Postgraduate | 4 (1.75) | 3 (2.56) | 1 (0.89) |
| **Marital Status** | | | |
| Single, never married | 63 (27.51) | 38 (32.48) | 25 (22.32) |
| Married or domestic partnership | 158 (69.00) | 72 (61.54) | 86 (76.79) |
| Divorced or separated | 6 (2.62) | 5 (4.27) | 1 (0.89) |
| Widowed | 2 (0.87) | 2 (1.71) | 0 (0.00) |
| **Health Insurance Status** | | | |
| Insured | 25 (10.92) | 15 (12.82) | 10 (8.93) |
| Uninsured | 204 (89.08) | 102 (87.18) | 102 (91.07) |
| **Income Level** | | | |
| Wealthy | 4 (1.75) | 3 (2.56) | 1 (0.89) |
| Living comfortably | 35 (15.28) | 22 (18.80) | 13 (11.61) |
| Just getting by | 120 (52.40) | 61 (52.14) | 59 (52.68) |
| Poor | 70 (30.57) | 31 (26.50) | 39 (34.82) |
| **Mean travel time to health center (min)** | 22.9 | 19.3 | 26.6 |
| **Have you ever received a Pap smear?** | | | |
| Yes | 203 (88.65) | 100 (85.47) | 103 (91.96) |
| No | 26 (11.35) | 17 (14.53) | 9 (8.04) |

Notes.
  [*]Some or complete.

and discrepancies were resolved through in-depth discussions (*Bradley, Curry & Devers, 2007*). Themes were derived from the detail-rich experiences of participants, which were discussed amongst research team members. Five main themes emerged from the data and were finalized through negotiated consensus (*Bradley, Curry & Devers, 2007*).

# RESULTS

## Quantitative sample description

The characteristics of both the urban (n = 117) and rural (n = 112) survey respondents are presented in Table 1. Study participants had a mean age of 28.1 (range 18–49). More than half of the participants completed some or all of secondary school (71%) in both the urban (85%) and rural (56%) populations. Significantly more women began or completed a university or postgraduate degree in the urban population (37%) than in the rural (13%). Most women who participated in the survey were married or in a domestic partnership

**Table 2  Time elapsed since last Pap test ($n = 203$)***.

| Time | Total ($n = 203$)* n (%) | Urban ($n = 100$) n (%) | Rural ($n = 103$) n (%) |
|---|---|---|---|
| Less than 3 months | 76 (37.44) | 40 (40.00) | 36 (34.95) |
| 3 to 6 months | 37 (18.23) | 15 (15.00) | 22 (21.36) |
| 6 months to 1 year | 50 (24.63) | 29 (29.00) | 21 (20.39) |
| 1 to 2 years | 25 (12.32) | 9 (9.00) | 16 (15.53) |
| 2 to 3 years | 5 (2.46) | 3 (3.00) | 2 (1.94) |
| 3 years or more | 9 (4.43) | 3 (3.00) | 6 (5.83) |
| Not sure | 1 (0.49) | 1 (1.00) | 0 (0.00) |

**Notes.**

*26 incomplete responses for survey item.

(69%) in both the urban (62%) and rural populations (77%). Most participants did not possess health insurance (89%). Although a similar fraction of urban (52%) and rural (53%) participants reported their household income level to be "just getting by," a larger percent of rural women (35%) described their income level to be "poor" compared to urban women (27%). On average, rural women reported traveling 7 min more to the health center than urban women. The majority of the women who participated in the study had previously received at least one Pap smear (89%) in both the urban (85%) and rural (92%) populations. Data indicate that 80% of women at the health centers had received a Pap smear within the past year, with 37% of women screened within the past 3 months (Table 2). In both the urban and rural populations, the majority of women had been screened within the past year (Table 2).

## Qualitative sample description

Our qualitative in-depth interview sample included 20 participants, 13 urban women and seven rural women. Though only 14 participants reported their specific age, the range was wide (19–46), with a mean of 32 years. Four recurrent themes representing determinants of knowledge and beliefs regarding cervical cancer screening arose from the interviews, as well as a fifth theme indicating participants' support of a potential HPV immunization program. Quantitative data from the surveyed population support these findings.

### Theme 1: Embarrassment associated with intimate nature of Pap smear and gender of exam provider

Participants indicated that some women are unlikely to have Pap smears as a result of the embarrassment and shame associated with the intimacy of the exam. Although some women have Pap smears to promote their health, others are too embarrassed to do so regularly. One woman described the shame she felt when receiving a Pap, and why she continues to regularly have the exam:

*Many [women] like the Pap because they want to be healthy, because health is life, but others do not because it causes them embarrassment…they say "the doctor will check me out down there, and I don't like that"…I let [my shame] go, but it still causes me*

**Table 3    Belief frequencies regarding Pap tests and HPV vaccine (n = 229).**

| Belief | Total (n = 229) | | Urban (n = 117) | | Rural (n = 112) | |
|---|---|---|---|---|---|---|
| | Agree n (%) | Disagree n (%) | Agree n (%) | Disagree n (%) | Agree n (%) | Disagree n (%) |
| Not comfortable receiving Pap from male provider | 161 (70.31) | 68 (29.69) | 78 (66.67) | 39 (33.33) | 83 (74.11) | 29 (25.89) |
| Teenage pap recipient more likely to have sex | 123 (53.71) | 106 (46.29) | 52 (44.44) | 65 (55.56) | 71 (63.39) | 41 (36.61) |
| Unsure of health benefits or purpose of Pap test | 59 (25.76) | 170 (74.24) | 35 (29.91) | 82 (70.09) | 24 (21.43) | 88 (78.57) |
| Takes a long time to receive results from Pap test | 145 (63.32) | 84 (36.68) | 78 (66.67) | 39 (33.33) | 67 (59.82) | 45 (40.18) |
| Likely to give daughter HPV vaccine if free | 224 (97.82) | 5 (2.18) | 114 (97.44) | 3 (2.56) | 110 (98.21) | 2 (1.79) |

*embarrassment, I still turn my face to the side. I feel this way because there are doctors touching me for the test, and although it shames me, I have to do it because it's for my health.*

*(Participant 16, urban, aged 37 years)*

Many women described their emotional discomfort with having a male provider performing their Pap smear. One woman described why she prefers having a Pap performed by a female provider:

*Emotionally, it's better with a woman [doctor] because with a man, [a Pap test] is more intimidating and you can't completely relax. While with a woman, she already knows what our parts are like, but the man only knows his parts and whatever he's studied of ours.*

*(Participant 14, urban, aged 24 years)*

Results from the quantitative survey (Table 3) support this finding, as more than 70% of women reported feeling uncomfortable receiving a Pap from a male clinician. Although the majority of women prefer to have a female clinician perform their exam, women spoke of having little choice over the gender of their provider at public health centers. The only women guaranteed to have a female clinician perform their Pap are those who attend private clinics, which provide services that are unaffordable for many women in the population.

### Theme 2: Association of Pap smears and cervical cancer with sexual promiscuity stigmatizes women with the disease

A recurrent theme was the belief that Pap smears and cervical cancer are associated with sexual promiscuity, and this association stigmatized and isolated women with cervical cancer in the region. Quantitative data (Table 3) illustrated a perceived relationship between cervical cancer screening and sexual promiscuity, as 54% of the population believed that teenage girls who receive Pap smears will be more likely to have sexual relations. This belief was more prominent among rural women (63%) than urban women (44%). Participant

18, a 26-year-old, urban woman spoke of Nicaragua as being a "conservative culture" with a "taboo about…information related to sexuality," especially in rural regions, where "they are more reserved."

Qualitative findings further developed this theme, linking the stigmatization of cervical cancer to the isolation experienced by women with the disease. Participant 15, a 46-year-old, urban woman described living with cervical cancer to be like "living alone." Another woman spoke of the isolation experienced by her aunt with cervical cancer, who did not tell her family about her cancer until five days before she died from the disease. The woman revealed the impact her aunt's experience had on her own perception of the importance cervical cancer screening:

> [My family] never thought any of these diseases existed until [my aunt's cervical cancer] happened and we suffered…what my aunt did of not talking about [her cervical cancer] was really bad because we were ignorant that she was suffering. For this reason it is good to discuss it and go get tested. I would say that women should get checked and have a Pap to detect disease or infection on time.
>
> (Participant 19, urban, aged 23 years)

The isolation experienced by women with cervical cancer can arise from the beliefs of an association existing between cervical cancer and HPV, a sexually transmitted infection that serves as an indicator of sexual promiscuity in the community. One participant depicted the discrimination faced by women with cervical cancer:

> [Cervical cancer] is a disease for which people take to discriminating against those who suffer from it, and take to talking about the people who have these problems. In reality it's a problem that we could all get, because even if a woman stays home protecting herself and taking care of herself, she doesn't know if her husband is out on the streets with someone else, so because of this many women stay quiet.
>
> (Participant 16, urban, aged 37 years)

This woman indicated that a root cause of the discrimination against women with cervical cancer is the association of the disease with sexual promiscuity. She suggests that a woman who is not sexually promiscuous but acquires HPV from her partner can still be blamed for developing cervical cancer. As a result of the association of cervical cancer with sexual promiscuity, women with the disease fear judgment from their peers and therefore do not speak of their experience with cancer.

### Theme 3: Knowledge of cervical cancer prevention limited to those who regularly attend health centers

Women reported their main source of knowledge regarding cervical cancer prevention to be *charlas,* or brief health workshops presented by providers at the health centers. Participant 14, a 24-year-old, urban woman stated that people in her community go specifically to health centers to receive "primary information" on these topics. Other women claimed that cervical cancer was only discussed in their community within the context of the health

**Table 4  Influence of variables on screening promoting-beliefs ($n = 229$).**

| | Impact on belief score: coefficient (confidence interval)[**,a,b] | |
|---|---|---|
| Variable name | Unadjusted | Adjusted |
| Urban vs. rural | −0.23 (−1.20, 0.73) | 0.63 (−0.38, 1.63) |
| Cervical cancer contact | 0.43 (−0.75, 1.61) | −0.13 (−1.17, 0.91) |
| Age | 0.03 (−0.03, 0.10) | 0.03 (−0.03, 0.09) |
| Knowledge | 0.81 (0.59, 1.02)[***] | 0.73 (0.47, 1.00)[***] |
| Education | | |
| No formal education | – | −0.98 (−6.55, 4.60) |
| Primary school[*] | – | 1.11 (−4.14, 6.35) |
| Secondary school[*] | – | 1.25 (−3.83, 6.34) |
| University[*] | – | 1.92 (−3.09, 6.93) |
| Postgraduate | – | REF |
| Income | | |
| Rich | – | 0.81 (−3.53, 5.14) |
| Living comfortably | – | REF |
| Just getting by | – | 0.20 (−1.05, 1.44) |
| Poor | – | −0.12 (−1.60, 1.37) |

**Notes.**

[*]Some or complete.

[**]95% CI.

[***]$p < 0.001$.

[a]The belief score is a summation of 22 survey questions that assessed participants' beliefs regarding physical and emotional discomfort associated with Pap smears, perceived safety of the exam, the amount of time it takes to receive the exam and results, the relationship between Pap smears and sexual promiscuity, frequency of screening, comfort receiving a Pap from a male clinician, and perceived likelihood of being affected by cervical cancer in the future.

[b]To determine the correlates of the belief score, unadjusted analyses were conducted using simple linear regressions and adjusted analysis using a multiple linear regression.

center. As a result, women who do not regularly seek care at these clinics have limited access to information regarding cervical cancer screening.

The results from the multiple linear regression (Table 4) indicate that greater knowledge of Pap smears, HPV, and cervical cancer was significantly associated with screening-promoting beliefs, which measures a woman's likelihood of being screened based on her responses to 22 belief questions (Appendix B). The multiple linear regression indicated that knowledge ($\beta = 0.73$, $p < 0.001$) was a significant predictor of belief score, even after adjusting for having had personal contact with cervical cancer, region of residence, age, education, and income. Region of residence, however, was not a significant predictor of screening-promoting beliefs.

The significance of knowledge in predicting women's beliefs was supported by qualitative data, which illustrated the self-efficacy and screening-promoting beliefs of women with knowledge of cervical cancer and screening practices:

*[The word "cancer"] is startling but I believe that yes, if they detect [cervical cancer] on time as I have my [regular Pap], I know that they can treat it with medicine. But if it is already too late, it is alarming that the consequence are already grave.*

*(Participant 14, urban, aged 24 years)*

In contrast, women who knew less about cervical cancer held abstract fatalistic views of the disease, rather than prevention-oriented beliefs:

*Well the truth is that I would not wish [cervical cancer] upon anyone because it is a disease that eats you from the inside. But if that is what God sends us, we have to accept it.*

*(Participant 4, rural, age not reported)*

Qualitative and quantitative data suggest that accurate knowledge of cervical cancer and screening practices may increase the likelihood of a woman holding screening-promoting beliefs. Although health centers are the primary source of information regarding cervical cancer and Pap smears, quantitative data indicate that 26% of women who were surveyed at health centers were unsure of the health benefits or purpose of a Pap smear (Table 3).

### Theme 4: Inconvenience of screenings at understaffed clinics increases patient wait time, limits time patients spend with clinicians, and delays Pap results

Women perceived screening to be inconvenient as they reported waiting for several hours before being seen by a clinician. Participants also indicated that clinicians were often too busy to spend a sufficient amount of time with each patient:

*For example, today I came in twice. The first time took two hours, and I couldn't get in because it turned out that the doctor was attending to pregnant women, and there were even more behind schedule… [which causes] the doctor to occasionally attend to patients hurriedly… right now there's one doctor. There are two that are missing. I don't know if they are on vacation or what, and that in part affects the general manner of the patients, because one gets restless and people get angry.*

*(Participant 17, urban, aged 47 years)*

Other participants explained that they were unlikely to receive regular Pap smears due to the amount of time it takes to be seen at the health center. One woman admitted that she should be having Pap smears more regularly, however was unable to spend hours at the health center:

*I have not had the time [to receive a Pap] because of work, but if I were to dedicate the day to see the doctor, I would get the results…*

*(Participant 20, urban, aged 38 years)*

In addition, over 63% of women reported having to wait a long period of time to receive the results from their Pap smear (Table 3). Women believed regular screenings were inconvenient due to long patient wait times, limited time with clinicians, and delayed Pap results.

### Theme 5: Support of potential HPV vaccination program

Participants expressed positive attitudes towards and acceptance of a potential HPV vaccination program. Although the majority of women interviewed had limited knowledge of the vaccine, every woman supported its use after learning of its ability to protect against cancer-causing strains of HPV. When speaking about the HPV vaccine, one woman stated:

 

*I've heard about it but I barely [had] any complete information…[the vaccine] would be really good because we are trying to fight [HPV] and that way we would be vaccinated against it.*

*(Participant 19, urban, aged 23 years)*

Nearly 98% of urban and rural women said that they would be likely or very likely to give their daughter the HPV vaccine within the next year if it were to be made free and available at the health centers (Table 3). Participant 7, a rural woman, stated that she would vaccinate her daughter as it would "save her" from HPV and cervical cancer. Another woman explained that even if she had to pay for the vaccine, she would try and provide it for her daughter:

*Even if [the vaccine] were expensive…health comes first.*

*(Participant 13, urban, aged 23 years)*

## CONCLUSIONS

Four recurrent themes indicate determinants of knowledge, beliefs, and attitudes regarding cervical cancer screening among urban and rural women of León. Our results suggest that women may be dissuaded from seeking regular screening due to embarrassment associated with the intimate nature of the exam, the association of Pap smears and cervical cancer with sexual promiscuity, and the inconvenience of long wait times in clinics. Women reported health centers were the primary source of information regarding cervical cancer and screening. A multiple linear regression indicated that greater knowledge of the disease was significantly associated with screening-promoting beliefs, even after adjusting for other relevant factors. Women were discouraged from seeking regular screenings as they believed that they would encounter long wait times to receive a Pap smear and test results. The fifth theme indicated overwhelming support for a potential HPV immunization program in Nicaragua.

Although the majority of participants preferred to have a female Pap provider, women reported having limited control over the gender of their provider at public health centers. This suggests that those with the financial means to receive Pap smears at private clinics with female providers may be more likely to seek regular screening than those limited to receiving care at public health centers. This finding builds upon previous literature that cites the exclusive use of public health services to be a contributing factor to low rates of cervical cancer screening in Rivas, Nicaragua (*Claeys et al., 2002*). In addition, our findings align with a review of qualitative studies conducted in Venezuela, Ecuador, Mexico, El Salvador, and Peru that suggest that poor service delivery, such as long wait time to receive a Pap, may also contribute to low rates of screening in Nicaragua (*Agurto et al., 2004*).

Our results indicate that there was no significant difference in the attitudes and beliefs regarding cervical cancer screening for urban and rural women, as region of residence was not a significant indicator of screening-promoting beliefs in the multiple linear regression. Thus, the data was not consistent with our hypothesis that urban women would hold stronger screening-promoting beliefs than those held by rural women. A

potential explanation of this result arises from our qualitative finding that most women in León learn about HPV and cervical cancer screening in health centers rather than from the general education system. Consequently, the increased amount of time urban women spend in school compared to rural women would not significantly affect their knowledge of the disease and screening. As knowledge is a significant predictor of holding screening-promoting beliefs, urban and rural women who have access to health centers may hold similar screening-promoting beliefs regardless of their region of residence.

Our findings must be interpreted in light of the limitation that a convenience sample was used in which participants were attendees of one of six health centers. We hypothesize that the convenience sample can, in part, explain our finding that the majority of both urban and rural women had received Pap tests within the past year. This finding stands in contrast to literature indicating that only 10% of women receive Pap smears at the nationally recommended frequency (*Pan American Health Organization, 2010*). We conjecture that the women who participated in the study accessed the health centers and were more likely to receive information regarding screening than their counterparts without regular access to the health centers, and thus, were more likely to be screened. Consequently, future studies could utilize a random sample to better understand factors that limit screening among women who do not regularly access the health centers, and how knowledge, beliefs, and screening frequency may vary between urban and rural populations. An additional limitation of our study was that we did not acquire an exact response rate for the quantitative sample; however, we were able to make an accurate estimate on the number of women who were eligible and willing to participate in the study, and calculated a precise final yield rate based upon the number of completed surveys (Fig. 1). It is also important to note the potential limited generalizability of our findings, which should be applied to other regions with caution. However, we note that our findings are applicable to both urban and rural populations of León, suggesting that they may be favorably applied to populations of varying demographics.

Despite its limitations, our research sheds light on several policy and practice implications that can reduce the incidence of cervical cancer in Nicaragua—the country with the highest cervical cancer mortality rate in Latin America (*Pulitzer Center, Public Radio International (PRI), 2008*). As knowledge was significantly associated with screening-promoting beliefs, future interventions should focus upon increasing access to information regarding cervical cancer prevention for women who do not regularly attend health centers. A program evaluation conducted in 2005 concluded that a partnership between local NGOs and the Nicaraguan Ministry of Health successfully delivered quality screening and health education programs through the use of mobile clinics in Nicaragua's rural North Atlantic Autonomous Region (*Howe et al., 2005*). The results from our investigation indicate the importance of expanding similar education programs throughout the country, as increased knowledge of cervical cancer is significantly associated with increased screening-promoting beliefs. Furthermore, our results indicate that over a quarter of women surveyed at the health centers were unsure of the health benefits of a Pap smear (Table 3), indicating that the educational capacity of the health centers can also be improved.

Though women described the stigma and embarrassment associated with screening, the majority of participants had received a Pap smear within the last year (Table 2). Qualitative data indicated that some women considered the potential health benefits of screening to outweigh the shame they associated with the exam. This finding aligns with the Health Belief Model (HBM), which indicates that perceived benefits and barriers to a health behavior, along with perceived susceptibility and severity of disease, affect an individual's action to prevent illness (*Jones et al., 2016*). Thus, knowledge of the benefits of screening encouraged women to receive Pap smears despite the negative attributes they associated with the exam. Consequently, we suggest that future education campaigns focus upon the health benefits of regular screening as an effective means of encouraging Pap smears in the region.

Furthermore, educational efforts should target additional components of the HBM, including perceived barriers to screening and perceived susceptibility to HPV and cervical cancer (*Jones et al., 2016*). Our findings indicate that beliefs such as embarrassment associated with the intimate nature of screening, as well as an association of Pap smears and cervical cancer with sexual promiscuity, discourage routine screening. By emphasizing the medical importance of screening and the prevalence of HPV and cervical cancer, educational efforts could mitigate perceived barriers to Pap smears and lessen the stigma surrounding HPV, cervical cancer, and screening.

This investigation is the first documented assessment of potential acceptance of an HPV immunization program in Nicaragua. Five Latin American countries with lower incidences of cervical cancer than Nicaragua—Mexico, Panama, Colombia, Peru, and Argentina—all include the HPV vaccine for girls in their national immunization programs (*Pulitzer Center, Public Radio International (PRI), 2008*; *Wigle, Coast & Watson-Jones, 2013*). Though the HPV vaccine can provide health benefits for both girls and boys, the predicted cost-effectiveness for vaccinating boys is limited when vaccination coverage for girls is over 50% (*Brisson et al., 2017*). Thus, national HPV immunization programs in Latin America have only targeted girls, and we focused our assessment on the potential acceptance of the vaccine for girls in León. Our results reveal overwhelming acceptance of an HPV immunization program, which has the potential to prevent nearly 70% of cases of cervical cancer (*Wigle, Coast & Watson-Jones, 2013*). Our results suggest that if funding were allocated to make the HPV vaccine accessible in Nicaragua, it would be well received and utilized to decrease the incidence of cervical cancer in the country.

## ACKNOWLEDGEMENTS

The authors thank Dr. William Ugarte of UNAN, León for his guidance throughout the project, as well as the staff at the health centers in which we conducted the investigation. We also acknowledge and appreciate the time participants dedicated to partake in the study. Thank you to Dr. Elizabeth Bradley and Dr. Amanda Brewster of the Yale Global Health Leadership Institute for their help with the data analysis and formulation of the manuscript.

# APPENDIX A. QUANTITATIVE QUESTIONNAIRE

1. Location of survey:
2. Date:
3. Time:
4. Survey identification number:
5. How old are you? (years completed)
6. Describe the zone in which you live:

   - Urban
   - Rural

7. Highest educational level you have completed:

   - No schooling
   - Pre-school
   - Some Primary School
   - Primary School Complete
   - Some Secondary School
   - Secondary School Complete
   - Some University
   - University Completed
   - Graduate Degree

8. How long does it take you to travel to the clinic (in minutes)?
9. Relationship/marital status:

   - Single, never married
   - Married or domestic partnership
   - Divorced/Separated
   - Widowed

10. Do you have private medical insurance?

    - Yes
    - No

11. What type of work best represents your current situation?

    - Government employee
    - Non-government employee
    - Self-employed
    - Student
    - Homemaker
    - Retired
    - Unemployed (able to work)
    - Unemployed (unable to work)
    - Don't want to answer

12. Describe your family-income level:

- Well off
- Living comfortably
- Just getting by
- Poor

13. Have you ever received sexual education?

- Yes
- No

14. Have you received education on the following themes? (Select all that apply)

- Puberty and reproduction
- Sex and sexually transmitted infections
- Transmission of HIV/AIDS
- Health relationships and communication
- None

15. Where have you received sexual education? (Select all that apply)

- School teacher
- Mother
- Father
- Sibling
- Son/daughter
- Other family members
- Friends
- Doctors/health clinics
- Internet
- Books/magazines
- Films/videos
- Church
- Other

16. Have you heard about a Papanicolau (Pap) test before today?

- Yes
- No

A Pap test is an exam of the anomalies of the cervix (below the uterus) that is performed during a pelvic exam.

17. Have you had a Pap test before?

- Yes
- No

18. How old were you when you had your first Pap test?
19. How long ago was your most recent Pap test?

- Less than 3 months
- 3 months to 6 months
- 6 months to 1 year
- 1 year to 2 years
- 2 years to 3 years
- 3 years or more
- Do not remember

20. How many days did it take to receive the results of your most recent Pap test?
21. Who performed your most recent Pap test?

   - Female doctor
   - Male doctor
   - Female nurse
   - Male nurse
   - Female student
   - Male student

22. How was the attitude of the person who performed your most recent Pap test? (Select all that apply)

   - Respectful
   - Professional
   - Friendly
   - Expert
   - Nice
   - Malicious
   - Disrespectful
   - Unprofessional
   - Rude
   - Inept

23. What was the result of your most recent Pap test?

   - Positive (abnormal result)
   - Negative (normal result)
   - Undetermined
   - Did not receive the results
   - Do not remember

24. How many children do you have?
25. Have you been pregnant during the past three years or currently?

   - Yes
   - No

26. Did you receive a Pap test during your most recent pregnancy?

    - Yes
    - No
    - I don't know

27. How likely are you to get a Pap test in the next 3 years?

    - Very unlikely
    - Somewhat unlikely
    - Somewhat likely
    - Very likely

28. What are reasons why you would be likely to get a Pap test in the next 3 years? (Select all that apply)

    - Tests for cervical cancer
    - Tests for HPV
    - Improves sexual health
    - Accessible at clinic
    - Clinic is easily reached
    - Promoted by doctors / nurses
    - Improves prenatal health
    - Recommended by friend
    - Required by physician
    - Other:

29. What are the requirements to have a Pap test? (Select all that apply)

    - Not menstruating
    - Not using a vaginal douche before exam
    - Not applying vaginal cream, medication, or suppository before exam
    - Not having intercourse three days before the exam
    - There are no requirements

30. Have you ever been denied a Pap test?

    - Yes
    - No

Do you agree or disagree with the following statements (#31–52)?

31. Pap tests are painful

    - Strongly agree
    - Somewhat agree
    - Somewhat disagree
    - Strongly disagree

32. Pap tests are safe

 - Strongly agree
 - Somewhat agree
 - Somewhat disagree
 - Strongly disagree

33. Pap tests can cause short-term discomfort

 - Strongly agree
 - Somewhat agree
 - Somewhat disagree
 - Strongly disagree

34. Pap tests can cause lasting health problems

 - Strongly agree
 - Somewhat agree
 - Somewhat disagree
 - Strongly disagree

35. Pap tests can increase likelihood of cervical cancer

 - Strongly agree
 - Somewhat agree
 - Somewhat disagree
 - Strongly disagree

36. You are unsure of the health benefits or purpose of Pap tests

 - Strongly agree
 - Somewhat agree
 - Somewhat disagree
 - Strongly disagree

37. Pap tests contradict with your religious beliefs.

 - Strongly agree
 - Somewhat agree
 - Somewhat disagree
 - Strongly disagree

38. Pap tests are too time-consuming

 - Strongly agree
 - Somewhat agree
 - Somewhat disagree
 - Strongly disagree

39. It takes a long time to receive the results of a Pap test

    - Strongly agree
    - Somewhat agree
    - Somewhat disagree
    - Strongly disagree

40. You would consult with your partner before receiving a Pap test

    - Strongly agree
    - Somewhat agree
    - Somewhat disagree
    - Strongly disagree

41. Your partner would have a problem with you receiving a Pap test

    - Strongly agree
    - Somewhat agree
    - Somewhat disagree
    - Strongly disagree

42. Pap tests are too physically invasive

    - Strongly agree
    - Somewhat agree
    - Somewhat disagree
    - Strongly disagree

43. Pap tests are too emotionally intimate

    - Strongly agree
    - Somewhat agree
    - Somewhat disagree
    - Strongly disagree

44. You do not feel comfortable receiving a Pap test from a male clinician

    - Strongly agree
    - Somewhat agree
    - Somewhat disagree
    - Strongly disagree

45. Pap tests are related to sexual promiscuity

    - Strongly agree
    - Somewhat agree
    - Somewhat disagree
    - Strongly disagree

46. You do not know where you can receive a Pap test

- Strongly agree
- Somewhat agree
- Somewhat disagree
- Strongly disagree

47. It would be difficult to receive a Pap test

- Strongly agree
- Somewhat agree
- Somewhat disagree
- Strongly disagree

48. The health center closest to you does not provide Pap tests

- Strongly agree
- Somewhat agree
- Somewhat disagree
- Strongly disagree

49. The health center where you would receive a Pap test is far away or hard to get to

- Strongly agree
- Somewhat agree
- Somewhat disagree
- Strongly disagree

50. You do not have sufficient information to decide if you should receive a Pap test

- Strongly agree
- Somewhat agree
- Somewhat disagree
- Strongly disagree

51. All adolescent girls should receive Pap tests

- Strongly agree
- Somewhat agree
- Somewhat disagree
- Strongly disagree

52. If a teenage girl receives a Pap test, she may be more likely to have sexual relations

- Strongly agree
- Somewhat agree
- Somewhat disagree
- Strongly disagree

53. How long (in years) after first having sex should women receive a Pap test?

54. How often should women receive a Pap test?

   - Less than 3 months
   - 3 months–6 months
   - 6 months–1 year
   - Every year
   - Every 3 years
   - Every 5 years
   - Every 10 years
   - Only once in a lifetime
   - Do not know

55. How much confidence do you have in Pap tests providing precise information about your health? (%)

56. Have you heard of human papillomavirus (HPV) before today?

   - Yes
   - No

57. Where have you received education about HPV? (Select all that apply)

   - School
   - Mother
   - Father
   - Brother/sister
   - Son/daughter
   - Other family members
   - Friends
   - Doctors/clinicians
   - Internet
   - Books/magazines
   - Films/videos
   - Church
   - No education about HPV
   - Other:

58. How probable do you believe it is for an individual to contract HPV during his or her lifetime?

   - Not at all probable
   - Slightly probable
   - Moderately probable
   - Very probable

59. Can HPV cause cervical cancer?

- Yes
- No
- Do not know

60. Is HPV is a sexually transmitted disease?

- Yes
- No
- Do not know

61. Do you think HPV infection can go away without treatment?

- Yes
- No
- Don't know

62. Can HPV be detected with a Pap test?

- Yes
- No
- Do not know

63. With whom would you feel comfortable talking about HPV? (Select all that apply)

- Someone at school
- Mother
- Father
- Brother/sister
- Son/daughter
- Other family members
- Friends
- Female doctor
- Male doctor
- Someone you just met
- Other:

64. Have you heard of cervical cancer before today?

- Yes
- No

65. Where have you received education about cervical cancer? (Select all that apply)

- Internet
- Television
- Film
- Newspaper
- Female doctors

- Male doctors
- Pamphlets
- School
- Family
- Government programs
- Programs of foreigners
- Other:
- Have not received education about cervical cancer

66. Can cervical cancer be a terminal illness (or can you die from cervical cancer)?

  - Yes
  - No
  - Do not know

67. Is there an effective method that significantly reduces the risk of cervical cancer?

  - Yes
  - No
  - Do not know

68. Have you had direct contact with cervical cancer (e.g., you, or one of your family members or friends, have had the disease)?

  - Yes
  - No
  - Do not know

69. How likely do you think an average woman is to get cervical cancer?

  - Not at all probable
  - Slightly probable
  - Moderately probable
  - Very probable

70. Do you think cervical cancer could affect you in the future?

  - Yes
  - No
  - Do not know

71. If you were to have cervical cancer, how much would it affect your life?

  - Not at all
  - A little
  - Moderately
  - A lot
  - Do not know

72. What do you think is the relationship between the following factors and cervical cancer? Classify each factor using a scale of 1 to 6, 1 meaning no relationship and 6 meaning a strong relationship.

   - Young Age
   - Genetic factors (occurrence of cervical cancer in close family member)
   - Human papillomavirus (HPV) infection
   - Human immunodeficiency virus (HIV) infection
   - Multiple sexual partners
   - Early sexual initiation
   - History of sexually transmitted infections
   - Alcohol abuse
   - Smoking
   - Miscarriages and abortions
   - A large number of pregnancies and childbirths
   - Early initiation of menstruation
   - Use of condoms
   - Hormonal contraception
   - Breastfeeding
   - Use of drugs or psychoactive substances
   - Using public swimming pools

73. Do you think that the following factors can reduce the risk of cervical cancer? Classify each factor using a scale of 1 to 6, 1 meaning no relationship and 6 meaning a strong relationship.

   - Diet rich in antioxidants
   - Regular physical exercise (more than daily activities)
   - Use of vitamin supplements
   - Proper long and relaxing sleep (minimum of 8 h per night)
   - Avoiding highly processed food
   - Avoiding genetically modified food
   - Weight loss
   - Refraining from casual sex

74. How effective do you think Pap smears are in preventing and detecting cervical cancer?

   - Slightly
   - Moderately
   - Very
   - Extremely
   - Do not know

75. With whom would you feel comfortable talking about cervical cancer? (Select all that apply)

   - Someone at school
   - Mother
   - Father
   - Brother/sister
   - Son/daughter
   - Other family members
   - Friends
   - Female doctor
   - Male doctor
   - Someone you just met
   - Other:

76. Have you heard about the vaccine against HPV and cervical cancer?

   - Yes
   - No

77. Have you ever heard about the HPV vaccine from any of these sources? (Select all that apply)

   - Health care provider
   - Friend or family member
   - Pamphlet or poster
   - Commercial or advertisement of a drug company
   - Television (not as an ad from a drug company; e.g., through a news story)
   - Radio
   - Internet
   - Newspaper
   - Program of foreigners
   - Government program
   - Other:
   - None of these options

78. Is the vaccine available in Nicaragua?

   - Yes
   - No
   - Do not know

79. Is the vaccine free of charge?

   - Yes
   - No
   - Do not know

The HPV vaccine is recommended for all girls aged 11–12. It protects against most genital warts and cervical cancer and requires 3 doses over a period of 6 months.

80. If you had a daughter, what would be reasons that might influence you to  NOT  have her get the HPV vaccine? (Select all that apply)

- Vaccine costs too much
- Not covered by insurance
- Concerns about vaccine safety
- May cause short term problems like fever or discomfort
- May cause lasting health problems
- Did not feel my daughter needed it
- She may be more likely to engage in sex
- She is too young to receive a vaccine for a sexually transmitted infection
- Have not been to doctor recently
- Religious reasons
- It is not effective in preventing cervical cancer
- Other:

81. If you had a daughter, what would be reasons that might influence you to HAVE her get the HPV vaccine? (Select all that apply)

- Effective in preventing cervical cancer
- Effective in preventing common strains of HPV
- Effective in preventing other sexually transmitted infections
- Effective in preventing genital warts
- Easy to access at clinic
- Free of cost/covered by health insurance
- Protects daughter when she becomes sexually active
- Recommended by doctor/nurse
- Recommended by friend/family member
- Recommended by ads/medical pamphlet/poster
- Other parents in the community are getting their daughters the HPV vaccine
- Other:

82. If the vaccine were completely free and available, how likely would you be to get the HPV vaccine for her in the next year?

- Definitely would not
- Probably would not
- Probably would
- Definitely would

83. How effective do you think the HPV vaccine is in preventing cervical cancer?

- Slightly effective
- Moderately effective
- Very effective
- Extremely effective

# APPENDIX B. SURVEY ITEMS INCLUDED IN THE BELIEF AND KNOWLEDGE SCORES

**Survey items included in belief score:**

1.   Pap tests are painful.
2.   Pap tests are safe.
3.   Pap tests can cause short-term discomfort.
4.   Pap tests can cause lasting health problems.
5.   Pap tests can increase likelihood of cervical cancer.
6.   Pap tests contradict with your religious beliefs.
7.   Pap tests are too time-consuming.
8.   It takes a long time to receive the results of a Pap test.
9.   Your partner would have a problem with you receiving a Pap test.
10.  Pap tests are too physically invasive.
11.  Pap tests are too emotionally intimate.
12.  You do not feel comfortable receiving a Pap test from a male clinician.
13.  Pap tests are related to sexual promiscuity.
14.  It would be difficult to receive a Pap test.
15.  All adolescent girls should receive Pap tests.
16.  If a teenage girl receives a Pap test, she may be more likely to have sexual relations.
17.  How long (in years) after first having sex should women receive a Pap test?
18.  How often should women receive a Pap test?
19.  How likely do you think an average woman is to get cervical cancer?
20.  Do you think cervical cancer could affect you in the future?
21.  If you were to have cervical cancer, how much would it affect your life?
22.  How effective do you think Pap smears are in preventing and detecting cervical cancer?

**Survey items included in knowledge score:**

1.   What are the requirements to have a Pap test?
2.   You do not know where you can receive a Pap test
3.   How likely is it to contract HPV?
4.   Can HPV cause cervical cancer?
5.   Is HPV is a sexually transmitted disease?
6.   Do you think HPV infection can go away without treatment?
7.   Can HPV be detected with a Pap test?
8.   Can cervical cancer be a terminal illness (or can you die from cervical cancer)?
9.   Is there an effective method that significantly reduces the risk of cervical cancer?

# APPENDIX C. DISCUSSION GUIDE FOR SEMI-STRUCTURED, QUALITATIVE INTERVIEWS

1. **Tell me about your experience with health care.**
   a. Do you usually come to this health center?
   b. Do you live in an urban or rural area?
2. **Tell me about your experience with sexual health education.**
   a. Where can women receive sexual health education?
   b. Do you want to receive more sexual health education?
   c. Do you think that adolescents today have sufficient information about sexual health?
   d. How could the sexual education of girls and women be improved?
3. **Have you ever had a Pap smear before?**
   a. What was the context?—Why? When? Who performed it? What was the purpose of the test?
   b. What are your concerns with the Pap test?
   c. Do you feel physically and emotionally comfortable receiving a Pap?
   d. Do you feel comfortable having a Pap performed by a male clinician? Why?
   e. How confident are you that a Pap provides precise information about your health?
   f. What is the relation between sexual promiscuity and the Pap test?
4. **What are your concerns regarding your sexual health?**
5. **Have you heard of human papillomavirus (HPV)?**
   a. What is your perceived risk of contracting the virus?
   b. What do you think is the general knowledge and attitude of the community towards HPV?
6. **Have you had experiences with cervical cancer before (personally, family, or friend affected by the disease)?**
   a. If you could change part of your/her experience with cancer, what would you change?
   b. What do you think when you hear the word cancer?
   c. How is the knowledge and attitude of the community toward cervical cancer?
   d. Who do you feel comfortable speaking with about HPV and cervical cancer?
   e. Have you spoken with your daughters and/or sons about these themes?
7. **How do you make decisions regarding your health?**
   a. With whom do you speak?
   b. What do you do to maintain your health and security against sexually transmitted infections?
   c. How worried are you about contracting a sexually transmitted infection?
8. **Have you heard of the HPV vaccine**
   a. If the vaccine were to be available and free, how likely are you to give the vaccine to your children? Why?

### Funding

Funding was provided by the Yale Global Health Studies Program, the Thomas C. Barry Travel Fellowship, the Class of 1960 Summer Traveling Fellowship, the Niebla Foundation Travel Fellowship, the Yale College Class of 2004 Travel Fellowship, the Davenport Richter Fellowship, and the Eduardo Braniff Fund at the Yale School of Public Health. The funders had no role in study design, data collection and analysis, decision to publish, or preparation of the manuscript.

### Grant Disclosures

The following grant information was disclosed by the authors:
Yale Global Health Studies Program.
Thomas C. Barry Travel Fellowship.
Class of 1960 Summer Traveling Fellowship.
Niebla Foundation Travel Fellowship.
Yale College Class of 2004 Travel Fellowship.
Davenport Richter Fellowship.
Yale School of Public Health.

### Competing Interests

The authors declare there are no competing interests.

### Author Contributions

- Hannah D. Rees conceived and designed the experiments, performed the experiments, analyzed the data, contributed reagents/materials/analysis tools, wrote the paper, prepared figures and/or tables, reviewed drafts of the paper.
- Alexandra R. Lombardo, Sara J. Meyers and Vishnu R. Muppala conceived and designed the experiments, performed the experiments, analyzed the data, contributed reagents/materials/analysis tools, reviewed drafts of the paper.
- Caroline G. Tangoren analyzed the data, contributed reagents/materials/analysis tools, prepared figures and/or tables, reviewed drafts of the paper.
- Linda M. Niccolai conceived and designed the experiments, contributed reagents/materials/analysis tools, reviewed drafts of the paper.

### Human Ethics

The following information was supplied relating to ethical approvals (i.e., approving body and any reference numbers):

The Yale University Human Subjects Committee granted ethical approval to conduct the study, as well as the Ethics Committee for Biomedical Investigations at the National Autonomous University of Nicaragua-León. IRB Protocol #1603017360.

### Data Availability

The raw data has been supplied as Supplemental Files.

## Supplemental Information

Supplemental information for this article can be found online at http://dx.doi.org/10.7717/peerj.3871#supplemental-information.

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
