# Peer review of "Knowledge and beliefs regarding cervical cancer screening and HPV vaccination among urban and rural women in León, Nicaragua"

_PeerJ, doi:10.7717/peerj.3871_

## Round 0.1 · original submission · Major Revisions

Your paper covers an important topic but there are some important issues to address, which mostly concern presentation of results and writing style.

Reviewer 1 ·

Basic reporting

Line 7: Please define VIA.

Lines 7-10: The authors should clarify that the type of work described in lines 7-10 has not previously been conducted in Nicaragua specifically because it has been conducted elsewhere.

Line 16: Unadjusted linear regression is typically referred to as simple linear regression, “single variable regression” should be replaced with “simple linear regression.”

Lines 18-19: What was the overall goal of in-depth, qualitative interviews?

Line 27: The 3rd theme does not sound like a barrier. Please revise to more clearly convey how this statement represents a potential barrier to screening.

Line 31-32: This sentence refers to an important finding (i.e., barriers to screening did not differ for urban and rural women); however, it is not mentioned in the Results section. This sentence would be more appropriately placed in the Results section.

Line 47-48: The source of the data reported in the following sentence should be referenced: “Vaccination against HPV-16 and -18 can prevent nearly 70% of cases of cervical cancer.”

Line 54: What is the cervical cancer mortality rate in Nicaragua?

Line 59-60: Please provide information on HPV and cervical cancer screening recommendations in Nicaragua. For whom (certain age groups?) is screening recommended in Nicaragua? With what frequency is screening recommended? Does the recommended frequency vary by age and risk?

Lines 73-74: Given that numerous studies have examined the acceptability of HPV vaccination programs, the authors should clarify that their statement in lines 73-74 refers to prior work in Nicaragua.

Line 91: When (i.e., months and year) was the study conducted?

Line 127: Does “diseases” refer to HPV and cervical cancer?

Line 133: Five weeks or six weeks as stated on line 92?

Line 141: Why were 92 surveys incomplete?

Line 188-189: The belief score clearly measures knowledge about and attitudes towards screening, but it is not defined in that way in the outcome variable section (lines 144-156), so it is unclear why the authors state that they examined correlates of knowledge and attitude (lines 188-189) in the quantitative data analysis section.

Line 189: Unadjusted linear regression is typically referred to as simple linear regression, “single variable regression” should be replaced with “simple linear regression”.
Line 190: What “variables of interest” were considered? Please clarify.

Line 247: According to Table 1, 52% of women were “just getting by” and 53% of rural women were “just getting by.” Please correct line 247 or Table 1 depending on which is currently incorrect.

Line 255: If aged 18-49 was an eligibility criterion, how were data on age missing?

All statements made about qualitative findings in the Results section should be limited to what women actually said. For example, it’s unclear whether the following statements reflect what women actually said during in-depth interviews or simply reflect the authors’ hypotheses about how women’s views about or attitudes toward screening might impact their willingness to undergo screening. If they reflect the latter, then they belong in the Conclusions section:

• Lines 287-289: “As these services are not affordable for many women in the population, the uncertainty of the gender of the provider may dissuade women from receiving Pap smears.”

• Lines 289-291: “The embarrassment felt by some women receiving a Pap smear may be exacerbated by a male Pap provider – effectively discouraging some women from regularly seeking cervical cancer screening.”

• Lines 334-335: “In addition, the perceived link between cervical cancer screening and sexuality could discourage women from seeking regular Pap smears.”

Line 337-338: The 3rd theme does not sound like a barrier. It should be revised to more clearly convey how this statement represents a potential barrier to screening.

Lines 367-375: Everything other than the following sentence between lines 367 and 375 seems to reflect the authors interpretation of their findings and belongs in the Conclusions section: “Furthermore, as shown in Table 2, 26% of women who were surveyed at 372 health centers were unsure of the health benefits or purpose of a Pap smear.”

Line 403: Delete “of” between “against” and “cancer-causing.”

Line 788: In question 37, change “you” to “your.”

Line 806: Should question 40 be changed to: You would consult with YOUR partner before receiving a Pap test?

Line 984: In question 67, change the final word from “method” to “cancer”.

Line 1159: In the table of survey items included in the belief score, in item 6, change “you” to “your.”

Line 1160: In the table of survey items included in knowledge score, in item 9, change the final word from “method” to “cancer”.

Experimental design

The research question is well-defined and the authors state how this study fills an identified gap in knowledge. However, the title of the manuscript, the first aim (lines 76-78), and the authors interpretation of findings from analyses addressing the first aim do not accurately reflect the quantitative data collected. Barriers to screening were measured via the belief score, which the authors state “measures a woman’s likelihood of being screened based on her responses to 22 belief questions” (lines 348-349). Although the beliefs measured likely have an effect on women’s decisions to undergo screening, the survey did not measure whether these beliefs actually serve as barriers to screening. For example, women were asked whether they agree or disagree with the statement that “Pap tests are painful.” Women were not asked whether they are unwilling to undergo screening because Pap tests are painful. Thus, it is unclear how the belief questions measure women’s likelihood of being screened. As such, the authors statement in line 348-349 requires justification.
Furthermore, the authors should either (1) address the discrepancy between the first aim of the study and justify their use of belief scores as the outcome of interest in quantitative analyses or (2) revise the Title, first aim of the study, and their interpretation of findings from analyses addressing the first aim to reflect the fact that barriers to screening were not actually measured in the survey.

Validity of the findings

Line 419-420: This statement is somewhat misleading. While the identified themes may pose barriers to screening, most of the quotes provided from in-depth interviews suggest that the identified themes reflect women’s views about or attitudes toward screening. Few of the quotes provided illustrate how a woman’s views/attitudes actually served as a barrier to screening. This statement should be modified to better reflect the findings presented.

Line 439-440: The following is not reported in the Results section: “Our results indicate that there was no significant difference in barriers to cervical cancer screening for urban and rural women.” All findings discussed in the Conclusions section should be reported first in the Results section.

Additional comments

I was able to open the raw data in Excel, but not in STATA. I may have done something wrong, but when I tried to open it, I got the following error:

. use C:\Users\Desktop\peerj-18174-Rees_STATAdata2.dta
file C:\Users\Desktop\peerj-18174-Rees_STATAdata2.dta not Stata format

Reviewer 2 ·

Basic reporting

This is a paper sharing results from a study in Nicaragua to understand knowledge, attitudes, beliefs about cervical cancer screening and HPV vaccination. Participants included a convenience sample of women recruited at 6 clinics (3 rural and 3 urban) in León, Nicaragua.

Experimental design

-This is a paper about beliefs and knowledge rather than “barriers” to screening. Please review and revise the aims, research question and focus throughout to reflect this.
- The methods section is very long, while it is clarify all methods there is some repetition. For example, it is stated several times that interviews were conducted in waiting areas.

Validity of the findings

no comment

Additional comments

Review of: Barriers to screening for HPV and cervical cancer among urban and rural women in León, Nicaragua (#18174)

This is a paper sharing results from a study in Nicaragua to understand knowledge, attitudes, beliefs about cervical cancer screening and HPV vaccination. Participants included a convenience sample of women recruited at 6 clinics (3 rural and 3 urban) in León, Nicaragua.

Major comments:
-This is a paper about beliefs and knowledge rather than “barriers” to screening. Please review and revise the focus throughout to reflect this.
-There is a lot of discussion in the results section, this should be moved to the discussion section.

Abstract:
-define the acronym VIA

Introduction:
-Line 59-61: You state that only 31.5% of women who are recommended to have a Pap test do have one each year. What are the screening recommendation for Paps in Nicaragua? This may be important to include. In the U.S. PAP screening is recommended every 3 years (not every year) for women age 21 to 65 years. On the PAHO Nicaragua cancer profile page (available here: http://www2.paho.org/hq/index.php?option=com_docman&task=doc_download&gid=23004&Itemid=270&lang=en) it states that screening is recommended every 3 years, so it wouldn’t be surprising that few women report having a Pap test in the past year.
-Line 77: No need to define the acronym HPV more than once in the manuscript.

Methods: The methods section is very long, while it is clarify all methods there is some repetition. For example, it is stated several times that interviews were conducted in waiting areas.
-Line 189: Single variable regression, please be specific and specify that this is linear regression.
-Please provide an explanation as to why so many surveys were incomplete and as to why those surveys were completely excluded.

Results:
-Line 346: “a multiple linear regression results” should be reworded.
-Most women report that that they have ever had a Pap test. How recent were those tests? This would be interesting to see since even though some women report that there are barriers to screening, they are still being tested.
-Move the discussion to the discussion section.

Questionnaires:
-Question 19 asks about how long ago participants had a Pap test. These results would be interesting to include in the paper.
-Question 23 doesn’t make sense as written. Pap tests do not test for HPV.
-Question 24 asks for number of children, it would be interesting to know more about how likely parents would be to vaccinate their children for HPV among those women that have children.
-Question 58 is not clear. Is this a result of the translation or is this exactly how the question was asked? Please provide some explanation for this question.
-There seems to be an error in question 67.
-What was the reason that you only asked about if people would be willing to vaccinate their daughters and not their sons (Questions 80+)? How many of the participants actually had daughters?

Tables: Add more descriptive title to tables and figure.
-Table 3, add footnote describing the statistics used so that results can be interpreted without going back to the methods section.

---

## Round 0.2 · Minor Revisions

There are numerous issues still to be addressed but these are relatively minor in nature.

Reviewer 1 ·

Basic reporting

1. I thank the authors for re-framing their manuscript from barriers to beliefs regarding cervical cancer screening so that their focus is more aligned with what was actually measured. However, the rationale for measuring beliefs is under-developed. One way the authors could strengthen their rationale is by discussing how women's beliefs regarding cervical cancer screening and HPV vaccination might influence (i.e., promote or inhibit) their uptake of these HPV and cervical cancer prevention strategies. This would make most sense prior to line 30 in the Abstract and line 103 in the Introduction.

2. It is unclear what the authors mean by “beliefs to promote or limit” in the Abstract (lines 34-45) and Introduction (line 107). If the rationale is further developed as suggested above, the authors could replace “to promote or limit” with “regarding.”

3. Line 40: Since the outcome was women’s belief scores, "and attitudes" should be deleted from line 40.

4. Line 41-42: This sentence should be deleted (“The outcome variable was the screening-promoting belief score.”). It would make more sense to define the outcome before (i.e., in line 39) describing the analysis. For example, “We assessed women’s beliefs regarding cervical cancer screening via a 22-item scale, with higher scores indicating screening-promoting beliefs.”

5. The authors’ suggested revision to lines 42-43 should be incorporated into the abstract: Twenty qualitative interviews, which sought to explain sociocultural dimensions of knowledge and attitudes indicated by our quantitative findings, were conducted with a sample of 13 urban and 7 rural women (aged 19-46).

6. I think the manuscript is greatly improved by shifting the focus of the manuscript to beliefs regarding cervical cancer screening. However, I think the manuscript requires a careful review to ensure that all aspects of the manuscript are up-to-date with respect to this shift in focus. For example, themes (1) and (2) sounds more like barriers than “knowledge and beliefs” regarding cervical cancer screening. Theme (5) also seems out of place with this description. Please revise (Abstract lines 46-55, each theme in the Results section, and lines 454-463 in the Conclusions section) to more clearly convey how these themes represent knowledge and beliefs. Perhaps attitudes towards cervical cancer screening would be an appropriate addition to the description of the themes identified?

7. Lines 55-56: Because this result is from the multiple linear regression analysis, this sentence should be presented with the quantitative findings above (i.e., before the presentation of qualitative findings).

8. Line 126: What is "reported acceptance"? The authors should consider deleting the word "reported" here.

9. Line 177-178: The latter half of this sentence describes how the belief score was analyzed (i.e., with respect to demographic characteristics and participant knowledge) and should be deleted as that information belongs in the Quantitative Data Analysis section.

10. Lines 183-187: While some items listed in Appendix B reflect positive statements (i.e., Pap tests are safe), most of the items listed in Appendix B reflect negative statements. Thus, it’s unclear how assigning a score of 1 if participants agreed with a negative statement would yield an overall score where higher values indicate screening-promoting beliefs. Please verify that the scoring is reported accurately and clarify if necessary.

11. Line 228: Knowledge should be deleted here since the women’s belief scores were used as the outcome, while knowledge of HPV and cervical cancer were explanatory variables of interest.

12. Lines 403-406: It’s great to see that these data have been added to the manuscript; however, these data would be more appropriately reported with the quantitative data at the beginning of the Results section.

Experimental design

The authors acknowledge that their sample is biased due to the fact that women were recruited from health centers, but their data suggesting that the majority of women in their sample are in compliance with national guidelines (lines 403-406: “80% of women at the health centers had received a Pap smear within the past year, with 37% of women screened within the past 3 months [Table 4]”) are in stark contrast to the estimate provided in the Abstract (line 30) and Introduction (lines 86-87): “only 10% of women receive Pap smears at the nationally recommended frequency.” This large discrepancy should be addressed in the manuscript.

Validity of the findings

Some of the authors qualitative findings directly highlight the mismatch between beliefs and actions (e.g., the quote from Participant 16 on lines 301-306). Because this study measured beliefs regarding cervical cancer screening rather than actual barriers to screening, it would be helpful if the authors further developed their discussion of how their findings could be used to develop interventions to promote cervical cancer screening and HPV vaccination.

Reviewer 2 ·

Basic reporting

This is a revised paper sharing results from a study in Nicaragua to understand knowledge, attitudes, beliefs about cervical cancer screening and HPV vaccination. This manuscript has improved since my prior reading, however there are still some issues that should be addressed prior to acceptance.

The language is clear. I suggest adding to your methods that the interviews and data collection occurred in Spanish and was translated after (or before) data analysis. I also suggest adding URLs to your references to allow readers to access your referenced documents. Lastly, the conclusion needs some editing to be more specific to the results from this particular study rather than sweeping statements- I have given some examples below.

Experimental design

The research question for phase 2 (qualitative) is not clearly articulated and different language is used to explain the purpose of this phase in different parts of the manuscript. Please see below.

Validity of the findings

Please see comments regarding the conclusion below.

Additional comments

-Abstract: It is redundant to say your simple linear regression analyses were unadjusted and multiple linear regression was adjusted.

Introduction:
-Line 70: Please provide a reference for “The vast majority of cases of cervical cancer are caused by persistent infection with specific strains of human papilloma virus (HPV).”

-Line 72: Please provide a reference for “In addition, screening through the use of cervical cytology is an extremely effective method to identify pre-cancerous lesions and prevent the development of cancer.”

Methods:
-Line 127-130: “The preliminary quantitative results influenced the creation of the open-ended discussion guide, which sought to elucidate sociocultural dimensions of knowledge and attitudes about HPV, cervical cancer, screening practices, and acceptance of HPV immunization that might further explain our quantitative findings. “ Is this sociocultural dimensions of knowledge, attitudes and acceptance, or personal/individual level dimensions? Later in Line 141-142 you state “The interviews thus placed a greater focus on understanding attitudes and beliefs towards HPV and cervical cancer.” This seems different than the aims previously stated. Then again later in Line 234-236 you state: “The purpose of Phase Two was to expound upon quantitative findings regarding gaps in knowledge and past experiences with Pap smears, HPV and cervical cancer. In addition, the interviews enabled a conversation regarding the views of women on a potential HPV vaccination program in Nicaragua.” These should be consistent and there is no reason for repetition. I suggest providing the aim/purpose clearly and concisely for phase 2 one time.

-Line 207-211: What was the reference level for your dummy variables?

Results:
-Line 274-276: “Fewer than half of the participants completed secondary school (46%) in both the urban (48%) and rural (44%) populations.” Looking at your table, that category is not completion of secondary school, but any secondary school. In addition, this doesn’t look like the accurate percentages – you can add categories together to stratify in this way.

-Table 3: Define “belief score” in a footnote.

-You present some data on health insurance. Please introduce the reader to the health insurance system in Nicaragua. Is any healthcare provided by the government? Are preventive health services like Pap Testing provided at a charge?

Conclusions:
-Thank you for the addition of Table 4. Table 4 demonstrates that almost all participants had had a Pap test within the past 3 years - just as the guidelines recommend. Therefore this sentence from line 455 (and other sentences from the conclusion) may not be relevant: “Our results suggest that women may be dissuaded from seeking regular screening due to embarrassment associated with the intimate nature of the exam as well as the association of Pap smears and cervical cancer with sexual promiscuity.” These contrasting results should be discussed. In addition did women specifically say that they were “dissuaded” from seeking regular screening? Or did they say that this was an issue they faced?

-Line 458: Did you measure health center attendance?

-Note: HPV vaccine is important for both males and females and may prevent cancer in both men and women. I suggest mentioning this in your discussion and providing an explanation as to why you focused on daughters rather than adolescents in general.

---

## Round 0.3 · accepted · Accept

Thank you for your careful attention to the extensive reviews.